# I'm Not Sure: Designing for Ambiguity in Visual Analytics

Stan Nowak*
School of Interactive Arts
and Technology
Simon Fraser University

Lyn Bartram†
School of Interactive Arts
and Technology
Simon Fraser University

## ABSTRACT

Ambiguity, the state in which alternative interpretations are plausible or even desirable, is an inexorable part of complex sensemaking. Its challenges are compounded when analysis involves risk, is constrained, and needs to be shared with others. We report on several studies with avalanche forecasters that illuminated these challenges and identified how visualization designs can better support ambiguity. Like many complex analysis domains, avalanche forecasting relies on highly heterogeneous and incomplete data where the relevance and meaning of such data is context-sensitive, dependant on the knowledge and experiences of the observer, and mediated by the complexities of communication and collaboration. In this paper, we characterize challenges of ambiguous interpretation emerging from *data*, *analytic processes*, and *collaboration and communication* and describe several management strategies for ambiguity. Our findings suggest several visual analytics design approaches that explicitly address ambiguity in complex sensemaking around risk.

**Index Terms:** Human-centered computing—Visualization—Visualization theory, concepts and paradigms

## 1 INTRODUCTION

Our work addresses the challenges of complex and collaborative sensemaking in risk management: in particular, the domain of avalanche forecasting responsible for analysis and prediction of snow avalanches endangering human life and infrastructure. As is the case with explaining and predicting other hazards (such as weather or natural disasters), avalanche forecasting involves the consideration and evaluation of alternative potential explanations that account for data [29] and the communication of these predictions to audiences widely varying in expertise. In this way, sensemaking is deeply about managing *ambiguity*, the state of multiple alternative meanings, and beyond simply accounting for missing information. Existing forecasting tools and procedures do not capture all the cognitive work forecasters do [51], motivating the design of better tools to support them. Because ambiguity is an essential component of their work environment, avalanche forecasters are an ideal study group for visual analytics interventions that explicitly target these challenges.

Visual analytics, "the science of analytical reasoning facilitated by interactive visual interfaces" [16], is well-suited to address the ambiguous sensemaking needs of avalanche forecasters. While visualization research has largely been devoted to quantified uncertainty or data uncertainty [8, 11, 17, 20, 33, 34, 47, 50, 63, 70, 73] researchers are now considering broader issues of uncertainty related to reasoning [85], such as the interpretation of implicit errors [55, 65], the importance of "hunches" in data interpretation [45] or the role of alternatives in visual analysis [46]. We add to this growing body of work in our exploration of the challenges of sensemaking un-

*e-mail: Snowak@sfu.ca
†e-mail: Lyn@sfu.ca

der ambiguity in risk analyses and the consequent implications for visualization designs.

In this paper we report work focused on two complementary threads of accommodating ambiguity in risk analysis and prediction. First, we seek a formative understanding of ambiguity in complex and critical sensemaking. Through a set of studies with Avalanche Canada, a public avalanche forecasting organization, we discovered the critical role ambiguity plays in sensemaking and its constant challenges for individual and collaborative analysis and communication. From these findings we characterized different sources of ambiguity and interpretative strategies, grouped into issues related to *data*, *analytic process*, and *collaboration and communication*.

Second, we describe how these findings informed initial visual analytics designs that explore better support for the challenges of ambiguous interpretation involving heterogeneous data-generating processes. We developed these tools in close and constant collaboration (participatory design) with forecasters. We then deployed them as design probes before redesigns were subsequently incorporated into daily practice, where we continue to observe their use. This ecological approach continues to surface challenges and affordances of supporting ambiguity in reasoning about risk in the collaborative and critical environment of forecasting. Our design findings highlight both the effective potential of visualizations and the caveats. Key issues are the importance of multiple levels of **data granularity**, appropriate **context**, the need for **analytic provenance**, and **enrichment** [3]: the ability to capture both data and insights throughout the process.

The key takeaway of our research is that ambiguity is distinct from data uncertainty, requiring solutions that go beyond reduction or removal. It is an essential component of sensemaking, but at the same time presents specific challenges for analysis, collaboration, and communication. We argue that ambiguity can and should be designed **for** and not **away** [60], that even simple design choices can serve to support or impede sensemaking involving ambiguity, and that there is a need for more explicit ambiguity support in visual analytics tools. In this paper, we contribute:

- Insights from 3 qualitative studies with avalanche forecasters surfacing issues of ambiguity in sensemaking;

- A characterization of sources and strategies for ambiguity in risk analysis and sensemaking; and

- A preliminary exploration of visual analytics design approaches to address ambiguity.

## 2 BACKGROUND

### 2.1 Public Avalanche Forecasting

Public avalanche forecasters assess avalanche hazards and communicate the associated risks to the public through daily bulletins. These natural disasters endanger the safety of humans and infrastructure and require careful professional assessment to inform risk management in mountainous avalanche-prone areas. Forecasters try to predict how present or future instabilities within the snowpack may react to natural triggers, such as the weight of new snow, or human triggers, such as the weight of a skier [54].

Avalanche forecasting is continuous and distributed across teams [51] of forecasters who monitor avalanche conditions over an entire winter season, iteratively updating their understanding with new information [54]. While many forecasters have the benefit of working in the field and directly observing avalanche conditions, public avalanche forecasters work remotely and rely heavily on field reports produced by other organizations [60]. In Canada, such reports are shared in the Canadian Avalanche Association's Industry Information Exchange (InfoEx) [25] by avalanche safety 'operators', such as those overseeing railway or transportation corridors, ski resorts, and helicopter skiing operations among others. While these data are structured and defined using formal measurement and reporting guidelines [2], they are gathered using a targeted sampling rather than a random sampling approach [54]. Operators actively seek instabilities in the snow. Consequently, forecasters have to glean enough context about this process to understand what such data mean (e.g. who reported it, where they went, what they saw, etc.).

Another challenge stems from the sparsity of data. For example, remote weather stations used to validate meteorological forecasts [60] are very sparsely distributed when compared to the variability and heterogeneity of mountain weather [48]. Forecasters mentally simulate the interactions of mountainous terrain and weathers systems and their effects on snowpack from limited data. This imaginative and speculative ability is a mark of competence and expertise in avalanche forecasting [1] as well as weather forecasting [67].

Forecasters formalize their judgements of avalanche hazards using a variety of qualitative measures such as a danger scale, likelihood scale, potential destructive size, as well as different avalanche types [76]. These assessments are then communicated to the public through daily bulletins that are supplemented with additional risk communications such as advice about how to avoid avalanche hazards. The public varies in levels of expertise and consequently varies in how they interpret even simple elements of bulletins such as danger scales [21, 75]. Public avalanche forecasters rely heavily on their knowledge, experience, and expert judgment to assess and communicate avalanche hazards. The challenges of complexity, varied interpretation, and uncertainty are similar to those involved in risk prediction and communication of other extreme weather events and natural disasters [7].

## 2.2 Sensemaking and Risk Prediction

Risk management work faces real-world time constraints, ill-defined goals, distributed tasks and responsibilities, uncertainty, and decision-making demands. The engineering of technological solutions to deal with these issues requires close consideration of the cognitive processes involved [30, 74]. Frequently in these domains, for example in weather forecasting [29], several targeted sensemaking strategies are employed. Generally, these involve the setting of expectations to direct attention to cues that can signal threats and a concurrent sensitivity to cues that deviate from these expectations [82].

### 2.2.1 Anticipatory Thinking

One example relevant to the forecasting of avalanches is *anticipatory thinking*: a functional form of mental preparation for potential risks including those that may be highly unlikely but could result in severe consequences [43]. Attention is actively managed and directed to subtle and context-sensitive cues that may signal threats. There are several types of anticipatory thinking. One, problem detection, describes the process by which observers first become aware of an issue that may require a course of action [40, 41]. The ability to detect problems depends on how rich an observer's understanding of relevant patterns to compare against data is. This "pattern matching" often involves monitoring multiple patterns or "frames" concurrently. Anticipatory thinking also involves "trajectory tracking", the extrap-

olation of trends into multiple alternative future scenarios as well as planning for them. The imagination, exploration, and planning for alternative scenarios is also known as mental simulation [38]. These processes are vulnerable to psychological factors or biases such as a tendency to explain away disconfirming evidence. However, studies with expert weather forecasters show such biases are countered through the active adoption of a skeptical stance in analysis [40].

## 2.3 Sensemaking and Ambiguity

Sensemaking — the process by which meaning is constructed based on available information and experience — is precipitated by information or events that violate expectations or are uncertain and ambiguous [52, 81]. It is characterized by complexity. Complexity involves dynamically evolving rules and interacting parts [28] where comprehensive understanding is intractable [37] due to the epistemological limitations of human observation [24]. These limitations mean that complexity is more effectively dealt with holistically rather than through mechanistic reduction to the sum of parts. Sensemaking addresses complexity and the concomitant uncertainties through the flexible construction of narratives [18] where informational *cues* help determine what is relevant and which narratives or explanations are coherent or acceptable to consider [12].

This "narrative mode" of thinking describes how signs, symbols, representations, and their relationships are tied together into coherent personal narratives authored by the observer [4]. A novel is merely ink until it is read by someone and the same applies to the analysis of data. Subplots and micro-narratives involving prior knowledge and personal experiences are involved in the reading and making sense of visualizations [61]. Just as a story involves competing narratives, so too, in general, does sensemaking. This is because sensemaking often starts with an existing explanation that is challenged by a viable alternative [39]. Sensemaking is thus more about resolving multiple potential meanings (ambiguity), rather than just accounting for missing or uncertain information.

## 2.4 Ambiguity in Visualization Research

Visualization research has a longstanding tradition of characterizing uncertainties relevant to the design of visual analytics systems [9, 49, 50, 79, 85]. Most visualization research has focused on data uncertainties, but many acknowledge the importance and role of interpretation and knowledge in uncertainty [20, 35, 49, 64, 85]. MacEachren discusses ambiguity through the lens of organizational decision-making describing it as a "*lack of an appropriate 'frame of' reference through which to interpret the information*" and describes equivocality as stemming from the diversity of possible interpretations [49]. Meanwhile, Boukhelifa et al. define ambiguity in terms of multiplicities in the relationship between entities and names in data as well as the differences in interpretation between collaborators [9]. Liu et al. present a framework for the exploration, interpretation, and management of alternatives in visual analytics [46]. They group alternatives into three types: cognitive (e.g. hypotheses, mental models, and interpretations), artifact (e.g. data, models, representations, or tools), and execution (e.g. methods, code, and parameters). Ambiguity is most closely related to their concept of cognitive alternatives. Researchers have discussed the challenges of ambiguity in natural language interfaces for visual analytic tools and developed dedicated mixed-initiative tools for user intent disambiguation [22, 31]. Most prominent in existing visualization research is the discussion of ambiguity in collaborative visual analytics where sharing of analysis is often incomplete, lacking context, and therefore ambiguous [27].

There is much more to analysis than what is explicit in data. Data are incomplete records of the phenomena they are intended to represent and require prior knowledge as well speculation. This is closely related to the notion of "implicit errors", which are errors inherent to a dataset but not explicitly represented within it [55, 65]. To better support sensemaking around implicit errors associated with

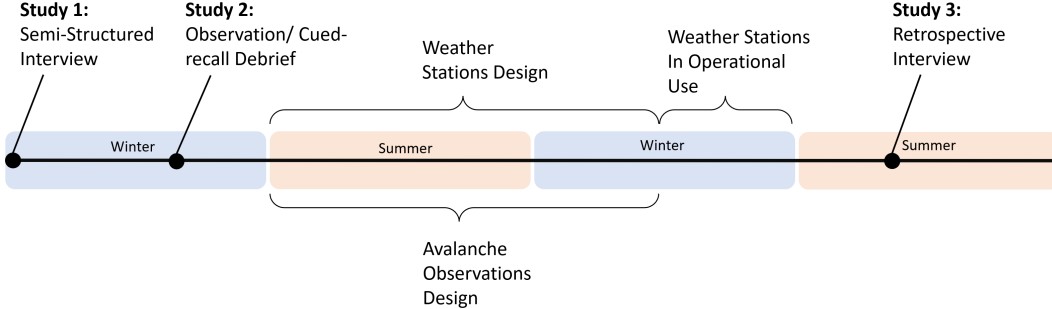

Figure 1: A timeline displaying the sequence in which studies were executed. Study 1 developed a formative understanding of avalanche forecasting challenges and workflows represented in a thematic code structure. This code structure was applied to observational data in Study 2 to refine understanding. Findings from these studies were used to inform the design of visualization prototypes used in Study 3.

infectious disease statistics, McCurdy et al. used structured annotations to help expert clinicians externalize knowledge about these errors [55]. In an application for archeological analyses, Panagiotidou et al. developed visualization tools that explicitly represented implicit errors [65]. Lin et al. use the term *data hunch* to describe "a person's knowledge about how representative data is of a phenomenon of interest" and how issues like credibility, inclusion and exclusion criteria, or directionality and magnitude of biases are considered in the analysis of data [45]. The authors outline a design space for externalizing data hunches.

## 3 APPROACH

We carried out 3 studies with forecasters at Avalanche Canada (Figure 1), a public avalanche forecasting organization. Our goal was to better understand the challenges of ambiguity in their sensemaking and to identify where visual analytics might help. We began with semi-structured interviews to understand how forecasters perceive and describe the challenges of their work (Study 1). We then conducted field observations of forecasters on site. Concurrently, we video-recorded forecasters' workstations and debriefed them about analytical reasoning involving the use of existing technologies (Study 2). This set of observations corroborated and enriched our understanding of the themes we identified in the interview study (Table 1). Subsequently, we implemented two fully functional visualization prototypes in collaboration with the avalanche forecasters and conducted retrospective interviews using these prototypes as design probes (Study 3). The purpose of this last study was to better understand how visual analytics interventions can address the challenges of ambiguity.

Studies 1 and 2 were conducted on-premises at Avalanche Canada while Study 3 was conducted remotely. In total, 12 avalanche forecasters participated in our studies (P1-P5 participated in Study 1, P2-P8 in Study 2, and P2-P6 / P9-P12 in Study 3). 10 were male and 2 were female, reflecting the gender balance of the organization and industry. The forecasters came from varied and mixed backgrounds. 8 had a background in professional mountain guiding, 3 in engineering, 2 in natural sciences, and 2 in business and communications.

We frame our findings according to issues of ambiguity dealing with **data**, **analytic process**, or **collaboration and communication**. **Data** are incomplete records of the phenomena they represent and require nuanced and varying interpretations depending on the needs and goals of analysis. Considering and evaluating alternative interpretations is an essential part of sensemaking: the **analytic process** of judging and adopting alternative interpretations presents potential analytic paths through data. These paths can be difficult to navigate as much of analysis is not explicitly captured. Finally, forecasters each hold unique perspectives and thus alternative interpretations

that need to be resolved. They rely on **communication** strategies that simplify complexity to retain clarity. This can obfuscate context and introduce ambiguities that their **collaborators** have to reason through. This structure arose from findings from our studies; we apply it in our discussion of the design implications for potential visual analytics solutions.

## 4 STUDY 1: FORECASTER WORK CHALLENGES

### 4.1 Procedure

We conducted semi-structured interviews with 5 professional avalanche forecasters on Avalanche Canada premises in Revelstoke, British Columbia. We asked about common work practices and challenges in avalanche forecasting, the role of data and evidence, the role of prior and tacit knowledge, issues of collaboration, and issues of uncertainty. Participants were asked questions like: "*Can you walk me through a typical forecasting day?*", "*What are the biggest challenges in your work?*", or "*What are some common uncertainties you deal with?*". The interviews were audio-recorded and then transcribed.

### 4.2 Analysis

Data were analyzed using thematic analysis [10]. Transcripts were concurrently segmented [23] and coded according to emergent themes by one coder. The codes were then refined in two passes. These themes were then grouped into thematic categories (Table 1). Inter-rater reliability was measured with one other coder who had a background in avalanche research and limited experience in qualitative research methods using a transcript sample representing 10 percent of all data [23]. Simple agreement for high-level themes was .89, Cohen's Kappa was .81, and Krippendorff's Alpha was .82. For the sub-themes, simple agreement was .75, Cohen's Kappa was .70, and Krippendorff's Alpha was .71.

### 4.3 Findings

#### 4.3.1 Data Challenges and Practices

*The data used in avalanche forecasting are uncertain, have ambiguous expressions or meanings, and have biases. These characteristics lead to ambiguity and a need to consider alternative interpretations beyond what is explicit in data.*

Forecasters told us one of their key challenges is the uncertainty involved in data sparsity or **missingness**. Data are often **explicitly** missing as is the case when remote sensors malfunction or fail to transmit. "[Weather stations] *that have good weather or wind information are even less, and then that's if they're even reporting [...]*" (P4). Missingness might also be **implicit** having to be inferred from the given situational context. "*In a large storm that closes*

*highways and grounds helicopters, it's very common the next day to not get any avalanche observations... but the weather and your personal experience very much suggests that there was going to be an avalanche cycle..."* (P1).

Forecasters rely on contextual information to understand how to appropriately interpret data following **circumstantial definitions**. Some of these contingencies are officially documented or ingrained within formal procedures, while others are only learned through extensive experience and knowledge. *"The [...] courses do quite a good job of standardizing those kinds of threshold amounts [...but] people who have spent a lot of time on the coast [...] may think a 30 centimeter storm doesn't really do very much..."* (P1).

Common to many classifications of the complex natural world, avalanche **classifications overlap** and are not mutually exclusive. Technically accurate hazard assessments might include several overlapping avalanche types resulting in overly complex public communications. Instead, forecasters try to choose a subset of avalanche types based on what may inform optimal risk mitigation strategies by the public. *"When you're modeling the natural world, you take shortcuts and there's simplifications[...] they don't occupy fully independent places [...] we sometimes have to have discussions about whether we want to be technically accurate, or whether we want to retain clarity [...] that starts to get quite complicated. [...] we look for ways to simplify..."* (P1).

The nuances of evidential reasoning and interpretation of data in avalanche forecasting also extend to the risk-based **conservative bias** common to forecasters. Some may be more or less conservative, and forecasters have to factor in such considerations when weighing evidence. *"[A]nother forecaster would have said something like: '[...]they always call that a little more than what it actually is.'[...]that] may influence me to say: Okay, well, maybe I should not necessarily discredit it, but I put less weight into it..."* (P3).

### 4.3.2 Analytic Processes and Reasoning

*Forecasters employ a variety of sensemaking strategies involving speculation and imagination. They integrate their prior knowledge, experiences, and contextual clues in data to synthesize understanding and explore risk implications.*

Forecasters synthesize, evaluate, and integrate information using a simulation technique they described as **mental projection**. It is a process of imagining oneself in the field to understand conditions and their risk implications. *"...that's a technique that a lot of people use to help forecast... kind of projecting yourself mentally, whether you close your eyes or you just have some kind of image of the kind of slopes, the kind of areas where the people are moving around [...] I think that experiential part there is really relevant to the process..."* (P1). This might involve mentally converting biases such as wind data from weather stations in windy locations. *"[T]here can actually not be that much wind in the park and you can have 60 kilometers an hour winds at that station. [...]taking an input and then adjusting it for myself..."* (P2). It might also involve simulating alternative future scenarios and their risk implications. *"If things are a little bit unusual, I [...] try and strip it down and build some kind of synthetic profile either in my mind, or sometimes even do it on the whiteboard [...] And then figure out the most likely, it's usually a set of scenarios..."* (P1).

Forecasters describe their work as bayesian-like because they are constantly updating their mental models with new information and **deliberately omitting** weak or redundant evidence. They reported having to **immerse** themselves in data over several days of their shift to build confidence in their sense of understanding. This often involves undirected explorations of general background information. *"...a day, you know, more likely two days to become fully sort of understanding of what's going on in your region [...] even if you can read it all in a day, it takes a little time for it to sort of percolate and*

*for you to understand what that means..."* (P1). To address identified gaps in understanding forecasters actively **seek contextual** sources of information. *"I'll [...] look for keywords like 'oh ya... skiing, like, steep terrain in the Alpine, up to 40 degrees and just exposed features. No problem.' That tells me that not much is going on. Yeah, people are confident..."* (P2). As they conduct their assessments, they iteratively **update** knowledge artifacts like the public bulletin to match their current understanding. *"I'm pretty iteratively making small changes in the forecast [...] I'll just move that right into the forecasts, put it there, save, and I go back to what I was doing..."* (P2).

Unlike forecasters, operators directly observe avalanche conditions in the field and thus have a richer understanding of the complexities involved. As a result, forecasters use subtle cues in data that can reveal the **subjective hunches** of operators to help them appropriately frame their understanding of avalanche conditions. *"'Okay, are these guys still concerned about this?' That's what really matters to me more so than like the really nuanced low-level data..."* (P2).

### 4.3.3 Collaborative Challenges and Practices

*Collaboration helps individual forecasters overcome the limitations of their own knowledge by drawing on the collective knowledge and experiences of their peers. At the same time, communicating the complexity of their assessments in simple terms is a constant challenge that creates ambiguities.*

Forecasters vary in knowledge and experience which likely contributes to some variations in interpretation. However, this diversity is seen as an advantage as, collectively, it addresses the gaps in understanding any single forecaster may have. *"[M]y experience may be different from you know... another forecaster's experience and I can learn from that person [...] there's those kinds of exchanges that happen..."* (P1). Forecasters share knowledge and solicit their peers' perspectives in daily **discussions**. *"At two o'clock, we have our pow-wow where we all kind of go through our hazards and our problems. [...] it's kind of like a peer review session..."* (P3).

**Professional exchanges** with partnering operations help avalanche forecasters enrich their understanding of how data are produced in a variety of operational contexts. *"[W]hether that's highways or ski hill, snowcat skiing, heli-skiing [...] there's variability between the individual operators... And the only way to really fully understand is to go and spend a bit of time with that operator. [...] We have professional exchanges go on..."* (P1). Forecasters also phone operators and **reach out directly** for clarification or if they are uncertain about how they should be thinking about conditions. *"[If I] am potentially missing something or I just don't feel comfortable [...] I'll start picking the phone up and trying to find people in the area that can provide more, more insight..."* (P3).

Collaboration allows forecasters to account for each other's knowledge gaps, at the same time, it presents challenges such as communication of analysis. Forecasting relies on the **continuity of analysis**. Shift-changes can disrupt this continuity and forecasters struggle with communicating relevant details as part of the hand-off process. *"[T]here's a lot of variability in different people and [...] what sort of information they leave [...] that's the first place I'll look [...] hoping that the [...] previous forecaster has left enough information to start that picture..."* (P3). To facilitate the hand-off process, forecasters **produce** knowledge artifacts like dedicated hand-off notes or detailed descriptions of snowpack stratigraphy. *"[Talking about hand-off notes] I am trying to take that ease and control that I have at day four or five [...] and I give that to the next person, so they don't feel like they have to do their process of discovery from ground zero essentially..."* (P2). This is seen as a separate and additional task often completed at the end of the day when forecasters are fatigued. This is why documentation used in support of hand-off

| S1 Theme | | S1 Sub-Theme | Definition | S2 Observed Evidence (O = Observation, C = CRD) |
|---|---|---|---|---|
| Data | Missing Info | Explicit | Missing information is explicitly represented in data. | |
| | | Implicit | Missing information must be inferred from the situational context. | O |
| | Data Representativeness | Classification Overlap | Classifications are often not independent or mutually exclusive. | O |
| | | Conservative Bias | Avalanche professionals are conservative when faced with uncertainty in the field or in data. | O |
| | | Circumstantial Definitions | Official definitions and unofficial practices for reporting data depend on the situational context. | O |
| Analytic Process | Analytic Practices | Subjective Hunches | Considering the behaviour, concerns, and hunches of others in the field to inform and guide analysis and interpretation. | C |
| | | Immersion | Forecasters spend several days forming a mental model through *undirected* review of contextual information. | C |
| | | Context-Seeking | *Directed* information search for supplementary contextual information. | C |
| | | Mental Projection | Forecasters assimilate information by imagining and mentally visualizing the interactions of avalanche conditions, weather, terrain, and people. | |
| | | Updating | Forecasters iterate over knowledge artifacts like their forecast as they conduct their analysis and update their own mental models. | C |
| | | Deliberate Omission | Forecasters manage information overload by ignoring certain data | C |
| | Analytic Challenges | Lack of Good Representations | Forecasters lament a lack of good visual representations to alleviate some cognitive effort. | C |
| | | Lowering Danger Ratings | It is challenging for forecasters to lower danger ratings as data reveal instability rather than stability. | |
| Collaboration and Communication | Collaborative Sensemaking Strategies | Continuity | Forecasting relies on the continuity of analysis and monitoring. Shift-changes disrupt this continuity. | O |
| | | Translating Analysis | Forecasters struggle with communicating complex conditions with simple clarity to the public. | O |
| | | Data Production | Forecasters facilitate collaborative work by producing hand-off notes and other internal knowledge artifacts. | O |
| | | Regular Discussions | Forecasters draw on each other's diverse knowledge through daily discussions. | O |
| | | Reaching out Directly | Forecasters call or email field operators for further information when faced with critical information gaps. | O |
| | | Professional Exchange | Forecasters work with other agencies and operators to gain a deeper understanding of the nuances of how data are produced and what they mean. | |

Table 1: Thematic codes developed in Study 1 (semi-structured interviews) and applied to Study 2 (field observations and cued-recall debrief). Thematic codes are organized and color-coded according to their relevance to different sources of ambiguity.

and collaboration is often incomplete.

Whether communicating to fellow forecasters or the public, capturing complexity and nuance in simple and understandable terms is a challenge. "*To simplify it* [...] *that's when you are kind of having to use your own best judgment...*" (P2). Forecasters must **translate** their understanding and cater it to an audience that varies in understanding and expertise. This often involves exploring alternative future scenarios, their implications, how an audience may interpret what the forecaster is saying, and subsequently choosing a simple communication strategy that comprehensively accounts for these alternatives. "*So instead of trying to write my forecasts like: 'oh, if we get 10 centimeters it will probably be okay, but if we get 20, then it'll probably come unglued'* [...] *It's like 'just watch for conditions to change as you increase with elevation* [...] *if it starts to feel stiff or slabby underneath your feet* [...] *use that terrain feature to go around it...*" (P2).

## 5 STUDY 2: OBSERVING AVALANCHE ANALYTICS

The purpose of Study 2 was to observe forecaster workplace behaviours and their use of technology. We sought a richer understanding of the challenges faced by forecasters and how visual analytics interventions might help.

### 5.1 Procedure

We conducted field observations on Avalanche Canada premises for a week., collecting field notes and audio recordings of daily discussions. At the same time, we gathered observations using cued-recall debrief (CRD), a situated recall method developed for use in complex decision-making contexts [62] and adapted for human-computer interaction [5]. 7 forecasters were observed in the field and 4 were debriefed using CRD. Camcorders positioned behind workstations in view of monitors and the desk surface captured recordings of forecaster's workday and their use of technology as well as artifacts such as hand-written notes. At regular intervals, video recordings were reviewed to identify timestamps where forecasters exhibited behaviours relevant to our research interests. At the end of the workday, recordings were played back to forecasters at marked timestamps, and forecasters were asked to explain their thought processes and actions. We asked questions like: "*Can you explain what you were doing and thinking here?*" These debrief interviews were video recorded and transcribed.

### 5.2 Analysis

We applied the thematic coding scheme developed in Study 1 to notes and transcripts in Study 2 (Table 1). This allowed us to compare what forecasters say and what they actually do. Thematic coding was applied by one coder in two passes.

### 5.3 Findings

#### 5.3.1 Analytic Tooling

Forecasters rely heavily on text tables and information from disparate web-based sources. They gather these resources in a map-based web portal that organizes hyperlinks to such resources spatially (Figure 2A). Data such as weather station telemetry representing meteorological conditions are investigated in a bottom-up manner. Telemetry from individual weather stations is viewed in a table format and iteratively synthesized into a holistic understanding of weather patterns. Similarly, professional field reports are generally viewed in text tables (Figure 3A). Forecasters scan down columns of tables to extract patterns and distributions from structured attributes such as avalanches sizes. At the same time, they read across rows of tables to extract details about individual reports to glean enough context to understand their significance. We observed forecasters repurposing web-browser features to accomplish simple analytic tasks. For instance, one forecaster opened several days of data in successive windows to investigate temporal patterns and make comparisons. This suggested forecasters could benefit from dedicated analytic tools to support such tasks. To our surprise, we found that the visualizations present in existing systems were seldom used. While it was clear the forecasters could benefit from dedicated analytic tools, the overwhelming use of text tables indicated this representational form held some comparative advantage in sensemaking.

#### 5.3.2 Talking About Data

*Organizational knowledge relevant to the nuanced interpretation of data is in large part oral tradition exchanged through the shared practice and environment of work.*

We observed several **discussions** that dealt with the topic of how to interpret particular reports. For instance, one discussion dealt with the interpretation of a report authored by an operator who was known to have a **conservative bias** and what the implications of this were for hazard assessments. In another discussion, a junior forecaster with a guiding background described how they are coming to understand the challenges of their new remote-work environment, noting the nature of what types of information may be missing. "*After having worked this job* [Avalanche Canada] ... *I sort of realize the big holes the operators leave in their writeups* [...] *because they are having face to face conversations... and maybe not putting that information into their writeup... saying this layer* [of snow] *does not*

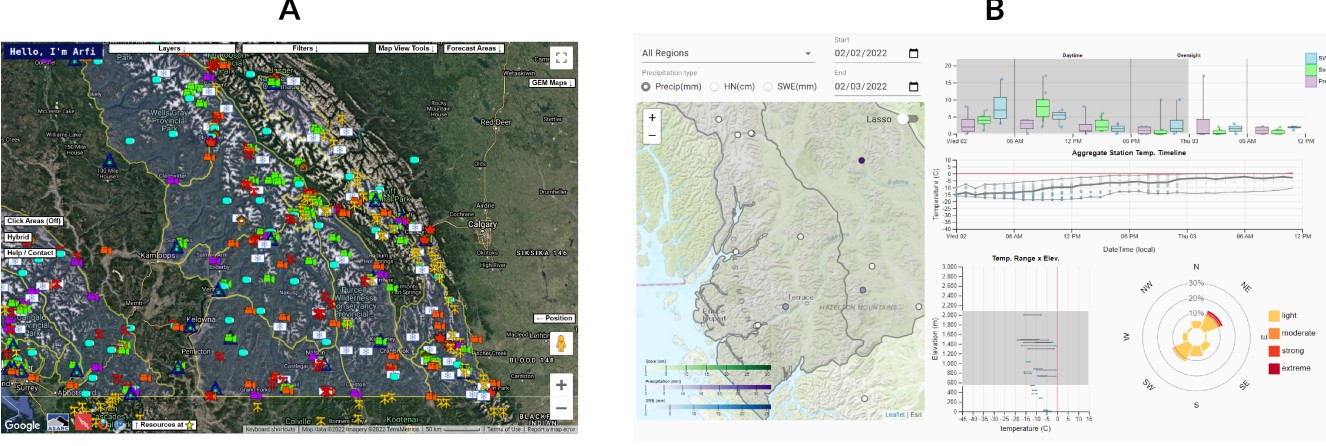

Figure 2: (A) Existing spatially oriented web portal linking to external weather station telemetry resources. Data from individual weather stations are commonly viewed in a table format and synthesized in a bottom-up manner. (B) WxObs visualization prototype showing numerical aggregates of weather station telemetry. Weather stations are viewed simultaneously using a conventional overview-first and top-down approach.

*exist in our area may not be helpful to them, but it really helps us here in this office...*" (P8). How classifications and circumstantial definitions are applied in hazard assessment and risk communication was also a frequent topic of conversation. "*I like* [X's] *point yesterday, wind slabs in the alpine are kind of like cornices that you find always... it is just a winter mountain hazard... it goes on the bulletin when it is elevated to more than normal caution...*" (P2).

### 5.3.3 Tacit Sensemaking and Analytic Processes

*Early sensemaking processes, particularly those involving personal experiences or trust, may be difficult to articulate out of context and consequently, share with others.*

When debriefing forecasters about their workday we found they relied on the **subjective hunches** of operators that they personally trusted and were more familiar with. This factored into how evidence was weighed and the confidence forecasters had in it. "*I feel good about who was about in the operation. So, I felt that the test was valid and valid information that I should be thinking about...*" (P3).

We also found forecasters exploring general contextual information to **immerse** themselves. They found it difficult to articulate how they were using the information, reflecting characteristics of early sensemaking processes [71]. "*It was just to give me an orientation to get my mental picture for forecasting* [...] *just a little bit of context... I don't know what that does for me exactly...*" (P4).

### 5.3.4 Collaboration and Knowledge Artifacts

*The bulletin serves as a knowledge artifact representing a forecasters' current understanding of avalanche conditions. The bulletin scaffolds analysis and guides information search, particularly during hand-off at shift changes. However, the reasons behind specific changes to the bulletin are not always explicitly captured leaving future collaborating forecasters to speculate about the reasoning that might have been involved.*

Forecasters don't just iterate over their own bulletin over the course of the day, they often carry forward the previous day's bulletin even if another forecaster wrote it. We observed how forecasters **update** it as they formulate their own new understanding. "*I import yesterday's forecast... and I tweak my forecast so it matches my now-cast...*" (P6). The specific reasons behind these updates are not made explicit, leaving the forecasters coming on shift to **seek contextual** information to speculatively reconstruct their coworker's evidential

reasoning process. "*...so I reviewed a few avalanches to understand what was driving those avalanches and why* [anonymized] *added that persistent slab problem again...*" (P6).

## 6 CO-DESIGNING VISUAL ANALYTIC SUPPORT

These findings guided us in developing visualization prototypes to support core forecasting tasks. We deployed these visualizations as design probes to examine how visual analytics interventions may aid in addressing challenges of ambiguity. The first prototype (WxObs) aggregates weather observations from remote weather stations in order to help forecasters validate the previous day's weather forecast as well as to monitor evolving weather systems in real-time. The second prototype (AvObs) uses field-reported avalanche observations produced by avalanche safety operations sharing data in the InfoEx. Avalanche observations are treated as key indicators of avalanche hazards in avalanche forecasting. We designed and developed both prototypes through several iterations from paper sketches to computational implementation in collaboration with avalanche forecasters. Both tools were evaluated using think-aloud protocol throughout the design process to explore how the tools support reasoning.

### 6.1 WxObs: Classic Design

Forecasters traditionally access weather station data through a spatially-linked web portal that redirects to external resources where data from individual weather stations are generally presented in text tables (Figure 2A). Forecasters use this information to synthesize patterns and distributions of various meteorological data such as precipitation totals, wind speeds, and temperatures. However, we found that their existing approach was challenged by the visual fragmentation and tediousness of accessing these disparate resources. We used a classic visual analytics linked and interactive multi-view design approach to streamline analysis and address this problem (Figure 2B).

We designed a conventional visual analytic display following Shneiderman's "Overview first, zoom and filter, then details on demand" visualization mantra [72]. Numerical aggregations of various weather stations telemetry across time and space were displayed in a variety of visualizations to provide forecasters with an "overview" of the data. Multiple "levels of detail" and "scales of resolution" of the data were captured across the display. All visualizations were linked together interactively supporting "brushing", "zooming", and "filtering" interactions across all corresponding displays. Individual marks visible in the spatial view allow tooltip interactions for

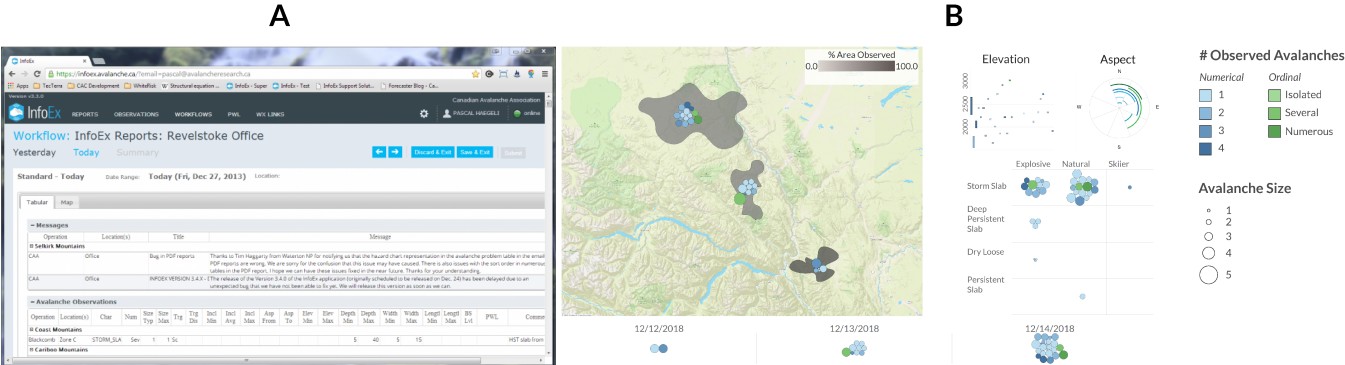

Figure 3: (A) Existing InfoEx interface displaying avalanche observation reports in a table format. Individual reports are read and analyzed in a bottom-up manner. (B) AvObs visualization prototype displaying avalanche observation reports using glyphs placed in a variety of visualization contexts. Individual reports are visible allowing critical contextual details to be discerned to inform understanding when there is a multiplicity of interpretations.

"details-on-demand".

## 6.2 AvObs: Breaking with Classics

Our second prototype, the AvObs tool (Figure 3B), uses daily field-reported avalanche observations shared by avalanche safety operators on the InfoEx platform. These tables are generally viewed in a tabular format. When we started designing this tool with the avalanche forecasters, we used classic visualization principles based on effectiveness and expressiveness [59] and common conventions such as using numerical aggregations. We found that even simple numerical aggregations like counts were problematic and inappropriate.

### 6.2.1 Disaggregated Data

We discovered several issues necessitating disaggregated views of data. First, the data have ambiguous expressions where the same data value may correspond to multiple meanings depending on context and the communicative intent of the author. Second, data are gathered using a targeted sampling approach rather than a random sampling approach. The data generating process is not uniform across the dataset and as a result, this challenges the methodological utility of aggregate measures.

### 6.2.2 Glyphs for Ambiguous Data

Forecasters wanted to see individual reports while at the same time being able to discern general patterns in the data. To address this design constraint we used glyphs with circle marks representing individual reports in a packed layout within a variety of visualization contexts. Circle marks were encoded using important structured data attributes within reports. The size of circles encoded typical avalanche size and the color encoded the number of observed avalanches. Two color maps were used to distinguish numerical and categorical values reflecting the need to preserve raw forms of data. Brushing and linking as well as tooltip interactions reveal contextual details allowing forecasters to discern how to interpret individual reports. This glyph-based approach operates at multiple scales of resolution allowing forecasters to visually aggregate data to discern patterns. Glyphs are known to support several visual aggregation operations such as summarizing data, detecting outliers, detecting trends, or segmenting data into clusters [78].

### 6.2.3 Desirable Difficulty

Early versions of the AvObs visualization prototype used bar charts that forecasters found difficult to interpret. They expressed concerns about visualizations giving them a **false sense of precision** and disarming the level of scrutiny forecasters usually apply to these data. We deliberately chose to use a visual design that we thought would break this sense of precision by introducing deliberate effort in decoding visualizations. We chose size and color as opposed to position which is commonly thought to be decoded more accurately [15] and, depending on the task, is often more perceptually salient [78]. In addition, combining visual features such as size and color is more difficult than using either alone [26]. In this way, we are explicitly violating the principle of perceptual effectiveness to provoke more deliberate consideration of the data, grounded in the concept of "desirable difficulty".

The benefits of introducing cognitive difficulties have been discussed in the context of geovisualization and risk-based decisions [13] and are well-documented in studies of human learning [84]. In visualization research, desirable difficulty has been framed as a trade-off between the cognitive efficiency derived from pre-attentive processing and improved learning through more active processing of information [32]. By reducing the *fluency* with which patterns in visualizations are read, more active and attentive processing of these patterns can stimulate "self-explanations" [14] where inferences about missing information are generated to fill in gaps or prior knowledge is integrated with new information to account for potential discrepancies. We conjecture our relatively more imprecise visualization design introduces visual complexity that induces additional effort, attention, and careful consideration of how perceived patterns should be interpreted. This is particularly important when ambiguity is a relevant consideration. By relying on quicker or more efficient information processing, one may be led to treat a visual display at face value and forego the consideration of alternative interpretations that may apply.

Beyond factors related to low-level perceptual processing, we conjecture that our chosen design serves as an effective metaphor for the messy nature of such data. Researchers have discussed how precise, easy-to-read, and minimalist designs can impart a sense of authority or objectivity [36] that may not always be warranted. The rhetorical force of visualizations to convince viewers that a clean visualization is an objective and perfectly truthful representation of the world can be detrimental when considering the messiness and complexity of many real-world data. Our deliberately messy design may serve as a reminder, much as tables do, that such data require additional scrutiny and interrogation from multiple perspectives.

# 7 STUDY 3: EXPLORING VISUALIZATIONS

## 7.1 Procedure

The visualization prototypes were evaluated using retrospective interviews. The avalanche observations prototype used simulated synthetic and historical data from past seasons and was never used operationally. The weather stations prototype used real-time data and was used operationally in the second half of the winter forecasting season. 7 forecasters had input on the design and development of prototypes while one simply commented on their experiences using them.

At the end of the forecasting season, we conducted semi-structured interviews asking forecasters to reflect on the prototypes, how they addressed the challenges of data, how they affected their work, and what needs remained unfulfilled. Interviews were conducted remotely using video conference tools. We used our prototypes as artifacts in the interview to prompt the forecaster's reflections. The interviews were video-recorded and transcribed. We summarize our key findings with quotes extracted from transcripts below

## 7.2 Findings

### 7.2.1 Many Possible Interpretations

The operational use of the WxObs prototype highlighted how analysis of weather station telemetry presents issues of **data uncertainty** that give rise to ambiguity. They are sparsely distributed relative to the large spatial areas they are used to represent [48] and they are subject to a variety of sensor and transmission errors caused by environmental factors. Presently, there is no comprehensive automated quality assurance procedure that accounts for all possible errors in the data [57]. Diagnosing errors and how individual weather stations come to represent broader weather patterns is a matter handled through the forecaster's judgment and interpretation. Forecasters normally use text tables to view each weather station's telemetry individually and progressively build up an understanding of weather patterns. This bottom-up approach stands in contrast to our top-down and overview-first visualization designs. Our visualization prototypes employed visualizations of aggregate measures, multiple granularities of data, interactions including brushing and filtering, and tooltips to view the details of individual weather station telemetry (Figure 2B). Our visualization prototype introduced a **new and unfamiliar analytic approach** that challenged forecasters. "*I've always looked at the data in a pretty disaggregated way* [...] *What I'm having to learn is to kind of let go of that, needing to see the disaggregated view first so that I can aggregate the data in my brain so to speak...*" (P12).

Similar challenges arose in the AvObs tool (Figure 3B). The human-reported avalanche observations follow reporting standards that, while structured, require a thorough understanding of context for interpretation. "*...the InfoEx system and the standards... they kind of define the box that we all work in* [...] *how you use them... context drives that. You might use a certain approach... data that are obviously within that general framework or box that we've created, but you might not use them exactly the same way...*" (P12). The same datum may be interpreted in a variety of ways and displays need to reveal the appropriate details for readers to discern what is appropriate.

### 7.2.2 The Need for Raw Data

Both data sources and prototype tools highlight a need for fluid interaction with underlying raw data. In the WxObs tool, many who are used to seeing **raw data** in a **tabular format** raised issues of **trust** as they could not apply the same visual scanning strategies to detect errors in data. "*[I]t largely stems from the trustworthiness of the data* [...addressing the use of spreadsheets] *I like things in their raw format just for my own sake* [...] *my own stamp of approval.* [...]

*I guess it's easy for my eyes to decode differences or irregularities. You should be able to visualize the data and get the same output. I don't know why. I just use tables...*" (P1). Others also used raw data tables but did so to **scaffold the learning** of data processing mechanics and the affordances of the visualizations as analytic tools. "*[...] having that* [raw data table] *side by side with the visualization helped me to interpret: Okay, what's the visualization trying to tell me here?*" (P4).

Similar issues surfaced with the AvObs tool. Early design iterations employing bar charts were seen as an impediment to sensemaking. Meanwhile, the glyph-based design was thought to hold more methodological utility as it more closely resembled and supported their mental model of how to analyze these data. "*I like seeing the individual events more than the aggregate... It seems like full of flaws and limitations to kind of summarize all the* [avalanche] *activity with one number...*" (P11). Despite our prototype using individual marks to represent each individual report, some forecasters still wanted the ability to see table-based displays. We speculate that this, similar to the WxObs tool, is due to issues of trust and learning how tacit analytic procedures associated with existing table-based views are or are not supported in the AvObs tool.

### 7.2.3 Forecaster Reflections

Forecasters who adopted the WxObs visualizations more readily in their work found the tool provided them with a richer and deeper understanding of meteorological phenomena than traditional data tables alone. Drawing a historical comparison to the role of computers in meteorology, forecasters view visualizations as a stepping stone in a **transitional phase** towards more data-driven modeling. "*[T]here was a transitional phase there where the computer was more an aid to help the forecaster make some initial assumptions... then the forecaster would tweak the forecast and actually write the forecast manually still... and now we're to the point where that really isn't the case...*" (P12).

Meanwhile, forecasters reported feeling satisfied with how the AvObs visualization prototype represented and supported their analytical processes. "*[The visualization] helps to smooth the data [...] and just at a glance [...] but it's not smoothing where I can't then [...] tease out nuances[...] I feel like it's really true to the data, which is a collection of individual points, kind of disparate points from across a forecasting region...*" (P2).

## 8 DISCUSSION

Throughout our 3 studies, we found that critical issues of ambiguity arise in three contexts: the data, the process of analysis, and the challenges of communicating both data and interpretation to both co-workers and the general public. We unpack the role of ambiguity, the concomitant challenges, and strategies used to deal with ambiguity in each of these contexts. Our findings highlight the need for more effective design interventions. We discuss each in turn.

### 8.1 Sources of Ambiguity

#### 8.1.1 Data

Ambiguity emerges from data because they are incomplete simplifications of the complex phenomena they represent. Ambiguity may be involved in the expression of data or how representative data are of phenomena of interest. Whether reasoning about multiple types of data uncertainty in weather station telemetry or what field-reported avalanche observations mean for avalanche conditions more broadly, forecasters use their knowledge, experience, and cues within the data to explore plausible explanations that account for what they see. Here, **provoking** alternative interpretations serves a productive purpose in analysis.

Forecasters try to **capture** relevant nuances of interpretation about specific data through daily discussions. Often this might serve to **disambiguate** meaning by providing an optimal or appropriate

framing for the data. For instance, the understanding that weather stations at windy locations will need adjustment when trying to understand broader wind patterns. We note that the forecasters' corpus of organizational knowledge is predominantly oral tradition exchanged in application to the immediate demands of work. Such a mechanism for knowledge exchange is vulnerable to information loss.

### 8.1.2 Analytic Process

Ambiguity both serves a productive purpose in analytic processes and presents challenges for the management and navigation of analyses. Alternative interpretations are explored as part of sensemaking often taking the form of alternative scenarios in risk analysis and risk prediction. Either through mental visualization or explicit sketches, forecasters **provoke** and imagine alternative scenarios to explore potential risks or explanations of data.

The judgments and analytic choices made during analysis represent alternative potential analytic paths through data. As forecasters weigh evidence and update their understanding of avalanche conditions, they iteratively adjust knowledge artifacts to match their understanding. However, the evidential reasoning process behind their judgments is often left **uncaptured** and may be difficult to reconstruct. This poses challenges for managing analysis as it may be unclear what work is completed and what remains to be done.

### 8.1.3 Collaboration and Communication

Forecasters each hold a unique perspective and interpretive lens presenting a form of ambiguity. Forecasters use strategies like regular discussions or hand-off notes to exchange knowledge and **disambiguate** how to interpret each other's assessments by **capturing** their reasoning processes. However, given the additional effort of this task and the difficulty in anticipating what may be relevant, such information is often not completed. This leaves forecasters having to speculate about their colleagues' reasoning processes.

Forecasters translate their own complex understanding of avalanche conditions in simple terms to ensure that members of the public, whether novice or expert, can apply appropriate risk-management strategies. In doing so, forecasters **mitigate** the risks of potential scenarios the public might encounter or the confusion that might result from overly technical communications. Quite often, this means **reconciling** alternatives. For instance, in a situation where two avalanche problem types require the same risk mitigation strategies, forecasters will use one of them and supplement any further guidance that might be necessary using plain and actionable language. The myriad of ways to communicate hazards presents its own form of ambiguity. Moreover, individual forecasters differ in how they judge avalanche hazards and apply assessments [44, 77].

## 8.2 Design Implications

### 8.2.1 When to Break the Rules

Conventional visualization design principles value precision-based visual variable *effectiveness* rankings as a basis for design decisions. However, as others have highlighted [6], this is an oversimplification of how visualizations are used. Visual pattern detection and visual thinking extend far beyond the precise extraction of singular values, and more importantly, displays that optimize for precision may have detrimental effects on other types of operations. With the need for close scrutiny of data and the potential for alternative interpretations, overly precise displays can give a false sense of precision and forfeit the perceived need for further scrutiny.

Our research has also highlighted that while the traditional 'overview first' mantra certainly has value in this application, it leaves a need for more fluid access and control to underlying raw data without overly onerous interactions. The properties of these data, like their ambiguous expressions or the varying data-generating processes, challenge conventional visualization approaches which can hide critical details that cue appropriate framing for data. While our designs shifted some focus to these cues, the need for bottom-up raw-data-driven processes was still highlighted in the feedback we received.

When dealing with heterogeneous and ambiguous data, designers should consider design approaches that best support the sensemaking processes involved rather than relying on conventional visualization mantras with a one size fits all approach. This reflects a broader need for improved guidance of how the affordances of visualization design can support the relevant cognitive processes needed for specific problem solving and sensemaking tasks. To do so, a characterization of what tasks can be supported by visualizations needs to move beyond what can be measured in lab experiments (e.g. low-level perceptual processes or decoding statistical properties of data). We suggest that a "macrocognitive" lens [42], one that values ecological validity and the complexities involved rather than strict control of variables, may help researchers identify such tasks.

### 8.2.2 Desirable Difficulty

Introducing cognitive difficulties in the context of visualization is thought to improve the memorability of insights [32]. Our research suggests that enabling or encouraging sensemaking around ambiguity is another beneficial outcome. There may be other benefits of introducing difficulties in visualization that remain to be identified.

### 8.2.3 Access to Raw Data Supports Sensemaking

Through our design study, we learned that visual displays of heterogeneous and ambiguous data should aim to reveal the relevant contextual details necessary to discern appropriate interpretations. Abstractions like numerical aggregations can occlude such details and impede sensemaking. Instead, we recommend designs such as unit visualizations that support visual aggregations or those showing the relevant granularity of data alongside numerical aggregations (in tables, for instance). This allows alternative interpretations to be provoked when trying to understand how data come to represent a phenomenon of interest. In addition, access to raw data can support the process of learning and adopting new analytic tools by revealing underlying data processing mechanics [3]. Hasty transitions to new analytic systems risk the loss of a host of implicit procedural knowledge that may not be supported by new approaches. This can cause issues of trust. Showing raw data alongside more abstracted views of the same data can aid comprehension of new tools and allow users to evaluate their affordances.

### 8.2.4 Capture Ambiguities Explicitly

We argue that design solutions need to extend beyond the representation of existing data. Managing an analysis with many contingencies and nuances of interpretation is difficult and is vulnerable to information loss, particularly when analysis is shared. To better serve the analysis at hand and to improve collaborative analysis, we suggest that the nuances of data interpretation should be captured explicitly during analysis. This would serve to characterize ambiguities through the externalization of relevant knowledge and the enrichment of data. We must take care these interventions remain lightweight and contextually anchored to avoid undue effort. We draw inspiration from the concept of "active reading", where knowledge generated during the process of reading is captured with external representations such as computationally-enabled markup and annotations [56]. Researchers have demonstrated that such techniques can be extended to analysis using visualizations [68, 80]. Annotations are a general-purpose technique that has been applied as a strategy to deal with ambiguity [9] as well as implicit errors [55]. This suggests annotations could be more specifically tailored and extended to address the challenges of ambiguity. Other forms of markup [3], including annotations, employed for the nuanced interpretation of data are often embedded in the ubiquitous spreadsheet,

perhaps the most widespread analytic tool. Tables are flexible and allow direct interaction with data which might explain why users often turn to them to support complex sensemaking. The affordances of tables are well-suited to deal with the challenges of ambiguity and may serve to guide the design of visual analytics systems in applications dealing with such challenges.

### 8.2.5 Externalizations Can Be Vague

Ambiguity is often the start of a sensemaking process. At such early stages, understanding may be inchoate and difficult to articulate, calling into question the utility of highly detailed capture mechanisms such as annotations. In collaborative analysis, it is difficult to anticipate the needs of others. Collaborators might only form an intuition about a problem that may be important for others to be aware of [40]. This is because the relevance of any such problem is context-sensitive [82]. Standardized protocols for sharing analysis often fail because designers of such protocols cannot adequately account for and predict all the unique information or complexity that might arise [66]. These considerations are important whether collaboration is with others or oneself at a future point in time.

There may be more simple capture mechanisms that can address the difficulties of articulating complexity. Passive capture mechanisms such as interaction logs provide one lightweight and context-sensitive solution. Interaction logs have been used to infer reasoning processes [19] and are frequently discussed as approaches for documenting analytic provenance [83]. Interaction logs, however, only show behaviours and are indirect indicators of reasoning processes. User-controlled markup may still be necessary to capture what is relevant. Researchers in clinical healthcare settings have supplemented hand-off protocols with vague metrics like gut feelings about a patient, time spent with a patient, or how medical equipment in a room has been moved around to take advantage of practitioners' shared work environment and culture [58]. We can take inspiration from this work. To capitalize on the shared digital working environment, simple markup such as tagging of data or representations may be all that is necessary to signify ambiguity. Tags may signify important pieces of evidence, how evidence is weighed and relates to assessments, or may simply serve to raise awareness of ambiguity and prevent it from being lost and risking potential misinterpretation. Forecasters can use their shared working environment to maintain context and capture ambiguities without having to precisely articulate them. Awareness of uncertainty is critical for ensuring trust in findings [69] and we argue the same applies to awareness of ambiguity.

### 8.2.6 Data Enrichment Requires Metadata Management

The use of more explicit data enrichment and ambiguity capture raises the question of how long captured data should persist as part of the working environment. Such markup may only be relevant for one working session and one individual. It might be relevant across several working days and for multiple collaborators. Or, it might take a more permanent form in a corpus of organizational knowledge. Designers should consider ways to control or account for the persistence of captured data.

Metadata created during analysis within a visual analytics system are bound to a representation rather than the underlying database. This raises questions about how such metadata may be queried, retrieved, or reused in contexts outside of the one they are created in and originally bound to. Designers need to consider how metadata can be reused and translated across analytic contexts.

### 8.2.7 Unstructured Metadata Require Schematization

Ad-hoc data enrichment and ambiguity capture pose some practical challenges when scaling. Annotations tend to produce large amounts of unstructured data that can be difficult to reuse. Such data require a schematization mechanism to make them tractable for future reuse.

Mechanisms for eliciting such data may be structured ahead of time, for example through survey-like questionnaires. Meta-data gathered at the time of elicitation, such as timestamps or application states [53], might also provide some structure. Alternatively, natural language processing approaches such as ontology-learning may lend themselves to schematizing such meta-data. However, we stress that the use of such algorithms should maintain transparency and give supervisory control to users. As we have learned in our design study, even simple statistical abstractions can obfuscate details paramount to reasoning about ambiguity. Further, highly complex technological solutions are more vulnerable to failure [82]. Consequently, the use of automation or algorithms should be carefully designed to make data processing transparent in support of human comprehension.

### 8.2.8 Baby and the Bathwater

Our experiences developing visualization prototypes for avalanche forecasters have highlighted the costs associated with introducing new analytic tools. The forecasters have developed visual reasoning strategies for interrogating data in table formats. Many of these procedures and processes are likely tacit and simply a natural habit that has been developed. When introducing new tools, even basic visualizations, there is a transitional period. A process of evaluating what capabilities are gained or supported, and which ones might not be supported needs to occur in practice. Until a thorough understanding of how a new analytic tool fits within the broader sensemaking toolkit, issues of trust will persist.

Computationally-enabled analytic tools are becoming ever-more sophisticated and complex. While there are real benefits to such powerful tools, designers need to consider the learning and unlearning of procedures associated with the adoption of new approaches. This is a common and obvious concern in the implementation of new systems. However, it is one that should be given more attention as it is often forgotten. This is particularly important in applications involving risk-based decision-making and time constraints where there are severe consequences for misinformed decisions.

## 9 LIMITATIONS

We note that while our first study had additional coders to test reliability, data from subsequent studies were analyzed by one coder only. Our comprehension of the challenges that forecasters face was incorporated in prototypes within our design study and the feedback forecasters provided throughout our close collaboration served as a form of validation of our understanding. This presents obvious limitations in the reliability of our findings. However, such challenges are common in the development of long-term, qualitative, and ethnographically inspired research aimed at deep domain understanding.

## 10 CONCLUSION

We have presented findings from a set of qualitative studies with public avalanche forecasters. Our research highlights that ambiguity presents challenges and unmet needs in critical and complex sensemaking. We propose a formative characterization of ambiguity across three levels of abstraction in analysis: *data, analytic process,* and *collaboration and communication.* The key lesson of our research is that ambiguity should be explicitly considered and designed for. While even simple visualization design choices can serve to enable or impede sensemaking around ambiguity, we argue for more targeted and explicit approaches. Our findings may inform future research and the design of tools in other complex risk-management domains such as extreme weather forecasting or the forecasting of other natural disasters. This work represents a preliminary attempt to characterize ambiguity and define a design space for visual analytics, but many questions remain unexplored. Further study is necessary to evaluate our existing and proposed

design solutions to more rigorously understand their impact and how they address the challenges of ambiguity.

## ACKNOWLEDGMENTS

Thanks to Avalanche Canada, the Vancouver Institute for Visual Analytics (VIVA), the Big Data Initiative at Simon Fraser University (SFU), the SFU Avalanche Research Program, and our reviewers for their thoughtful feedback. This work was supported by Mitacs through the Mitacs Accelerate program and the Natural Sciences and Engineering Research Council Industry Research Chair in Avalanche Risk Management (grant no. IRC/5155322016), with industry support from Canadian Pacific Railway, HeliCat Canada, Canadian Avalanche Association, and Mike Wiegele Helicopter Skiing.

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
