# OpenReview forum: "I’m Not Sure: Designing for Ambiguity in Visual Analytics"
_graphicsinterface.org/Graphics_Interface/2022/Conference — GI 2022_

### Official Review · Reviewer_t7iT · 2022-01-15
**Strong paper opening new theory on the role of ambiguity in practice**

**Rating:** 8
**Confidence:** 4

**Review:**

This is a well-written manuscript presenting a series of three studies of domain experts involved in analyzing real-time data to make critical decisions about avalanche safety. The three studies are an interview study, an observation of the analytics work, and a follow up on the deployment of two prototypes presented as design probes. The central discussion of the paper is around the role of ambiguity in the workflow of avalanche prediction, but I found the concepts presented to give me new ideas about the importance of recognizing ambiguity in data and process in visualization. It also raised interesting points around provenance of data - even at the micro level. For example, I found it fascinating that some of the avalanche analysts would trust observations from some field observers more than others. A traditional visualization would most certainly hide this type of metadata. The role of raw data tables in analysis was important, and even after using the presented prototypes, some of the analysts wanted access to the data tables. The paper positions raw data access as a way to bridge adoption of new systems, as well as a way to provide assurances to experts who want to check disagreggated data to better understand the uncertanties, missing data, etc. that may be obfuscated in higher level visualizations. Issues such as visualization providing a false sense of precision are not new but this work reaffirms the danger and shows that domain experts also are aware of it. Overall, this is an interesting work presenting a case study in a specific domain that collects, clarifies, and affirms a lot of knowledge about the role of ambiguity in analysis.

Minor issues:
- missing period first paragraph of Section 2
- Fig 2 caption has some odd spacing
- backwards quote marks in some places
- ellipsis with two dots instead of three in places
- I personally found the use of bolded words to be too much. Some were useful, but I think it could be scaled back.
- Section 5.3.3 the quote "At two o'clock 230" -> what does this mean?
- Section heading capitalization is not consistent (5.3) and in Section 4 Study 3 is referenced as study 3

---

### Official Review · Reviewer_d481 · 2022-01-15
**Interesting qualitative studies on managing ambiguities in the visual analysis process**

**Rating:** 7
**Confidence:** 3

**Review:**

This paper conducted three qualitative studies with public avalanche forecasters to identify and characterize the challenges they face. In particular, the authors characterize the challenges of ambiguous interpretation during sensemaking and present a preliminary design probe for visual analytic solutions to address such ambiguities. The first two studies examined the challenges of the forecasters as well as their workplace behaviors and their use of technology. Then the authors designed some prototypes for supporting forecasting tasks as design probes to understand how visual analytic solutions may address the challenges of ambiguity in the avalanche forecasting domain.

Strengths:
- Three different in-depth studies reveal important insights that would not only be useful for the specific domain but also for other domains where challenges of ambiguity exist in the visual analysis process.
- Overall, the qualitative studies are done in a rigorous way
- Visualization design probes were helpful in understanding the possible problem space and the possible solution

Cons:
- some parts of the paper seem too verbose and difficult to follow without appropriate headings. For example, the Design implications sec. would probably be more readable with explicit headings for each implication/point.
- Fig 3 is difficult to follow without a sufficient explanation of what each visual component is depicting in the caption. Annotating different visualizations in that figure and then explaining them explicitly from the caption would help.
- There are few papers that examined ambiguities in visual analysis that arise while using natural language input. The authors may consider adding them to the literature review. Here are a couple of papers:

- https://dl.acm.org/doi/10.1145/2807442.2807478
-https://ieeexplore.ieee.org/stamp/stamp.jsp?arnumber=8019833

---

### Official Review · Reviewer_R658 · 2022-01-16
**Substantial, useful work that suffers from a lack of structure and clarity**

**Rating:** 6
**Confidence:** 4

**Review:**

Summary
The authors study ambiguity in high-risk sensemaking - here, in avalanche forecasting - through three data sources: 1) semi-structured interviews with 5 experts, 2) observations on avalanche analytics workplace and data, and 3) feedback with two prototype visualizations. This data is analyzed and presents several themes that can inform visualization design involving ambiguity.

Review
This paper reports on substantial work that could provide a useful contribution, but the value seems to be buried in the paper. I believe a major revision to clarify the final takeaways, summarize the results with more structure and skimmability, and link the results to the final takeaways would result in a much stronger paper and clearer contribution. However, because the methods are sound and some clarification can be made in minor revision, I think this paper has the potential to be accepted. In short: It could be a great paper with a lot of writing effort, or an acceptable contribution with some smaller revisions.

Positives
The introduction and background sections are quite strong. They lay out some of the theoretical background and appear to be well-grounded in the literature (however, I have a passing awareness on current information visualization research). The methods seem reasonable, although a few more details would help us trust the results more (see below). There is a lot of information here that could add to the conversation surrounding ambiguity in visualizations design.

Primary concern: buried outcomes
There are many data sources, with interesting quotes sprinkled throughout. I think there is some nice information here. However, the three study sections run together as a long stream of quotes, making it difficult to link the data to the constructs in the paper (ambiguity, etc.). The reader easily loses the story from all of this data strung together, and when the final outcomes are given, they are vague and high-level, and I find it difficult (but not impossible) to figure out how to incorporate the results into a design or follow-up research project. To remedy this, I think the authors should try to structure the results into more concrete sections and subsections, and rework the themes to be complete sentences/claims supported by the results, rather than categories of quotes. (Admittedly, this can be a stylistic choice, but I think it would help here.) Figure 2 might be further enriched with details. With better structure, the themes can then be linked to outcomes that are more concrete and actionable, with more details about what designers and researchers can do in follow-up work.

Minor concerns (easily fixed in minor revision):
 - Section 2 interrupts the flow, I wonder if it can be incorporated into related work (or create a “Background” section that covers the domain and the related work)
 - make a more explicit rationale why Avalanche Responders are the right people to study for this problem in the introduction (this should be easy and only one sentence at most)
 - the reader needs more information about the coding process - what methods/methodologies are used? How is the coder trained? What sample of the transcript (how much was used for the inter-rater reliability?) Citations would be very helpful here.
 - quotation formatting needs work (use parentheses, careful bout sentence endings), if the themes are given more structure, they can be more skimmable or organized with headings/subheadings
 - Figure 1 needs a longer caption to help explain it. Careful about typos (“semi-structure interviews” or “semi-structured interviews”?)

---

### Decision · Program_Chairs · 2022-01-18

Accept